# The bone phenotype associated with cherubism is independent of Caspase-1-dependent inflammasome activation in the mouse

**Badre-Victor Rabhi[1], Sylvie Thomasseau[1], Xavier Decrouy[2], Martine Cohen-Solal[1,3], Marcel Deckert[4], Amélie E. Coudert[1,5☯*], François Brial[1,3☯*]**

**1** BIOSCAR, Inserm U1132, Université Paris Cité, Paris, France, **2** Plateforme Imagerie, IMRB - Inserm U955, UPEC, Créteil, France, **3** UFR de Médecine, Université Paris Cité, Paris, France, **4** MICROCAN, C3M, Nice, France, **5** UFR d'Odontologie, Université Paris Cité, Paris, France

☯ These authors contributed equally to this work.
* amelie.coudert@inserm.fr (AEC); francois.brial@u-paris.fr (FB)

## Abstract

Cherubism is a rare genetic disorder caused by *SH3BP2* mutations. This sterile autoinflammatory disease is characterized by jaw osteolysis, in which bone tissue is replaced by multinucleated giant cells containing fibrous tissue. The cherubism mouse model (*Sh3bp2* KI) is characterized by systemic bone loss as well as inflammatory phenotypes induced and maintained by TNFα. IL-1β, produced by the NRLP3 inflammasome through recruitment of Caspase-1, is involved in the development of sterile autoinflammatory disease. We previously reported a cherubism patient with elevated serum IL-1β, and cherubism mice also have elevated serum IL-1β levels. Thus, we wanted to disentangle the role of IL-1β in cherubism. To that end, we deleted *Caspase-1* in *Sh3bp2* KI mice to tamp down IL-1β production. However, deleting *Caspase-1* did not rescue the systemic bone and inflammatory phenotypes.

## Introduction

Cherubism (OMIM #118400) is a rare pediatric bone disorder [1] characterized by jaw osteolysis in which bone is replaced by fibrous tissue containing osteoclast-like multinucleated giant cells [1]. The disease was first described by Jones in 1933 [2]. It was called cherubism because of the swollen cheeks that make the patients look like the cherubs of the Sistine Chapel [2]. Cherubism is an autosomal dominant disorder. Most cases are caused by gain-of-function mutations in the *SH3BP2* gene [3]. Recently, mutations in another causative gene, *OGFRL1*, have been described [4]. Osteolysis of the jaw usually occurs between 2 and 5 years of age and spontaneously regresses after puberty [1]. The bone lesions are bilateral, symmetrical and may involve all the bones of the jaw up to the orbit floor, but without pain [1]. Most case reports of cherubism have described the craniofacial phenotype without considering the rest of the skeleton. Recently, however, we reported systemic bone loss in a young cherubism patient [5]. Although various molecules have been proposed for the treatment of cherubism, there is

**Data availability statement:** All relevant data are within the paper and its Supporting Information files.

**Funding:** This work was supported by grants from the BIOSCAR, INSERM U1132. B.R. was a recipient of fellowship from MENRT (Ministère de la Recherche). F.B. was a recipient of a Université Paris Cité Starting Grant.

**Competing interests:** The authors have declared that no competing interests exist.

currently no standardized therapeutic protocol for the disorder [6], other than surgery. What triggers the specific anatomic location and time course has not been identified. Cherubism with its benign osteofibrous lesions is defined as a sterile autoinflammatory disease. To gain insights into the pathogenesis of cherubism, knock-in (KI) mouse models have been generated [7,8]. The cherubism phenotype in *Sh3bp2* KI mice is characterized by systemic inflammatory and bone-loss phenotypes, mainly reported in 10-week-old animals. At the biochemical level, mouse cherubism is characterized by increased secretion of TNFα [8], which is responsible for the development and maintenance of the bone and inflammatory phenotypes. In humans, the inflammatory status of cherubism remains unclear, although elevated serum concentrations of TNFα have been reported [5]. However, an anti-TNFα treatment, etanercept, proved to be ineffective [9]. The serum levels of other inflammatory cytokines are not as well documented. For IL-1β, the published data are scarce but suggest elevated serum levels in mice [10,11]. In addition, the cherubism patient with the systemic bone-loss phenotype that we described also had an elevated serum IL-1β level, more than three times the upper limit of normal [5].

IL-1α and IL-1β are the founding members of the 11-member IL-1 family. IL-1β plays a central role in several sterile autoinflammatory diseases [12,13]. IL-1β activation has been shown to be regulated by inflammasome complexes [14]. Several inflammasome complexes have been described, but NLRP3 (NLR family pyrin domain-containing 3) has been the most studied, particularly as a major producer of IL-1β [15]. The NLRP3 inflammasome consists of three main components: the Nod-like receptor molecular scaffold NLRP3, the adaptor molecule called apoptosis-associated speck-like protein-containing CARD (ASC), and pro-Caspase-1 [15]. NLRP3 inflammasome activation involves a 2-step process. The first step is cell priming leading to NFκB activation and the production of pro-IL-1β. The second step consists of assembly of the inflammasome machinery, including recruitment of ASC and pro-Caspase-1 by NLRP3. Caspase-1 is then activated and cleaves pro-IL-1β to IL-1β.

This study focuses on the consequences of *Caspase-1* deletion leading to IL-1β alterations in the pathogenesis of cherubism. Our goal was to demonstrate that by deleting *Caspase-1* and thereby impairing NLRP3 inflammasome function, a normal bone phenotype could be restored in *Sh3bp2* KI mice.

## Materials and methods

### Animals

The *Sh3bp2* Knock-In (*Sh3bp2* KI) and *Caspase-1* knock-out (*Cas1* KO) mouse models were previously generated and described elsewhere [7,16]. Both mutants are maintained on a C57BL/6 background. Heterozygous mice from the two lines were crossed to generate the *Sh3bp2 KI;Cas1 KO* double mutant line. Hereafter, for simplicity, we use *Sh3bp2 WT* for *Sh3bp2$^{+/+}$*, *Sh3bp2 KI* for *Sh3bp2$^{G418R/G418R}$*, *Sh3bp2 Het* for *Sh3bp2$^{G418R/+}$*, *Cas1 Het* for *Cas1$^{+/-}$*, *Cas1 WT* for *Cas1$^{+/+}$*, and *Cas1 KO* for *Cas1$^{-/-}$*. *Sh3bp2 KI,Cas1 KO* mice were obtained by crossing heterozygous mice (*Sh3bp2* Het;*Cas1* Het) and were analyzed at 10 weeks. The heterozygous mice are viable and fertile, and were mated to generate all the genotypes used in the study: wild type (*Sh3bp2 WT;Cas1 WT*), defined as control mice; knock-in mice (*Sh3bp2 KI;Cas1 WT*); knock-out mice (*Sh3bp2 WT;Cas1 KO*); and double mutant (*Sh3bp2 KI;Cas1 KO*) mice. Identification of animals of interest was performed by PCR genotyping using the primers listed in Table 1, according to the following PCR conditions: 5 minutes at 94 °C; 35 cycles of 94 °C for 30 seconds, 60 °C for 1 minute, 72 °C for 1 minute; and a final step at 72 °C for 10 minutes. The amplicons were then analyzed by electrophoresis on a 2% agarose gel.

The mice were bred under controlled conditions at 22 °C with a 12-hour dark/light cycle. All experiments were authorized and approved by the Ministry of Higher

**Table 1. PCR genotyping primer list.**

| Primer name | Sequence | Amplicon size |
|---|---|---|
| SH3BP2 KI Erf1 (Reverse) | 5'-ACA-GAG-ATA-GCC-CCT-CAC-CCT-GAAG-3' | WT 150 bp |
| SH3BP2 KI Ef1 (Forward) | 5'-CGT-GTG-GAG-CAG-TGG-AAG-TAG-3' | KI 258 bp |
| ICE5 (Reverse) | 5'-GAG-ACA-TAT-AAG-GGA-GAA-GGG-3' | WT 500 bp |
| ICE3 (Forward) | 5'-ATG-GCA-CAC-CAC-AGA-TAT-CGG-3' | KO 300 bp |
| ICENEO (Forward) | 5'-TGC-TAA-AGC-GCA-TGC-TCC-AGA-CTG-3' | |

Education and Research and the local Ethics Committee n°9 (authorization APAFIS #39914-2022112517318664).

On the day of sacrifice, the mice were anesthetized by intraperitoneal injection of 100 mg/kg ketamine and 10 mg/kg xylazine. The mice were photographed, weighed and measured, and bone density was also measured at 10 weeks, and then the mice were euthanized. Blood was collected by cardiac puncture, and 40–150 µL of serum was collected after centrifugation at 3500 rpm for 10 minutes. Bones (femur, tibia, skull and vertebrae) were dissected out and cleaned of muscle. Several organs were collected for various analyses, including the spleen and liver. Livers were fixed in AntigenFix (MM France, France) for 24 hours, and bones were fixed in 70% ethanol.

## Bone density analysis (DXA)

Whole mice were scanned by DXA using an X-ray energy of 40 kV, 0.20 mA and an exposure of 2.90 s (UltraFocus DXA, Faxitron®). Regions of interest (ROIs) for whole body, femur and lumbar vertebrae were drawn around the bone image, from which bone mineral content (BMC, mg) and bone mineral density (BMD, g/cm$^2$) measurements were obtained.

## MicroCT analysis

Skulls, femurs and vertebrae were fixed in 70% ethanol for 5 days and scanned with a Skyscan 1272 (Bruker, Belgium) at the U1132 imaging facility (Paris, France). The samples were scanned by microCT with an X-ray energy of 70 kV and 100 mA with an Al 0.5 mm filter and an exposure time of 1000 ms. The angular step between image acquisitions was 0.5°, and each acquired image was an average of 4 frames. A spatial resolution of 6 µm was used for femurs and vertebrae and 12 µm for skulls. The obtained images were reconstructed to perform the bone microarchitecture analyses according to published guidelines [17]. Three-dimensional reconstruction images were obtained and analyzed using NRecon and CTAn software (Bruker, Belgium), respectively. Analysis of the mandibular bone volume/tissue volume (BV/TV) was performed as previously described in the cherubism mouse model by Yoshimoto *et al.* [18]. The region of analysis included 20 sections of furcation area below the first mandibular molar. For skulls, the analysis was performed as previously described in the cherubism mouse model by Ueki *et al.* and others [8,11,18–23], but we developed a semi-automated protocol. Briefly, the volume of interest (VOI) selection was performed using DataViewer software (Bruker, Belgium) at the level of the calvarial sutures. For image analysis, we developed a specific macro to perform the calvarial erosion analysis using FiJi/ImageJ software (version 2.15.0) [24]. The calvarial erosion analysis procedure begins with a folder containing a set of images of a mouse calvaria. The images from this folder are imported as an image sequence. A z-projection is performed in a single plane. A square selection of 6 mm per side is automatically generated. The position of this square is adjusted by the user. The threshold is checked manually. The area is measured. The threshold value and the measured area are saved in a csv file for later statistical analysis.

## Histomorphometry

After fixation, femurs were embedded in methyl methacrylate for histomorphometric analysis as previously described [25]. Five-micron tissue sections were obtained and TRAP activity was detected as previously described [25], and the number of osteoclasts on the bone surface was counted in the metaphyseal area below the growth plate. Sections were counterstained with 0.03% aniline blue and the TRAP+ cells were counted using a bright-field microscope as recommended [26].

## Histology, staining and quantification

Liver tissues were fixed in AntigenFix for 24 hours and then embedded in paraffin. Five-micron tissue sections were cut and stained with Hematoxylin and Eosin (H&E, Sigma). For data acquisition, images of liver sections were captured with a Zeiss Axioscan 7 at 10× (NA 0.45), using Zen 3.7 software (Zeiss, Jena, Germany). The inflammatory lesions were evaluated by machine learning. The process was as follows: First, we trained the Ilastik software (version 1.40 post1) on selected sections stained with H&E to classify inflammatory clusters, normal tissue and background. Then, an entirely new batch of sections was classified using the trained Ilastik software. A pixel classification model was generated using Ilastik software [27]. The generated pixel segmentation image was used in a macro for Fiji/ImageJ software (13) to detect and measure the clusters.

## Inflammatory cytokine levels measured by Milliplex®

Serum cytokine concentrations (TNFα, IL-1β, IL-18) were analyzed as previously described by using a Milliplex® kit (Cat # MPXMMAG-70K-02, Millipore, Paris) on a Luminex platform provided by Centre d'Histologie, d'Imagerie et de Cytométrie (CHIC) du Centre de Recherche des Cordeliers (CRC, Paris).

## Osteoclast culture from splenocytes

*In vitro* osteoclastogenesis from splenocytes was performed as previously described [25]. Briefly, splenocytes were isolated from 10-week-old mice of each group (Sh3bp2 WT;Cas1 WT, Sh3bp2 WT;Cas1 KO, Sh3bp2 KI;Cas1 WT, Sh3bp2 KI;Cas1 KO). A cell suspension was obtained using a 70-μm nylon mesh cell strainer (Falcon, Dutscher, France). After centrifugation (1300 rpm, 8 min, 4 °C), a red blood cell lysis step (5 min, room temperature) was performed and the remaining cells were washed with αMEM (ThermoFisher, France) and centrifuged again. The number of cells in the suspension was counted. Cells were seeded in αMEM (ThermoFisher, France) containing 10% fetal bovine serum (Hyclone, ThermoFisher, France), 1% penicillin/streptomycin (ThermoFisher, France) and 1% L-glutamine (ThermoFisher, France) in 1- or 8-well chamber slides (Lab-Tek®, Dutscher, France) at $2 \times 10^6$ cells/mL. Cultures were fed with fresh medium supplemented with M-CSF (25 ng/mL, Preprotech, France) and with RANKL (30 ng/mL, Preprotech, France) every 3 days for 14 days.

## TRAP activity staining

After 14 days of culture, cells were washed three times with phosphate-buffered saline, fixed with 4% paraformaldehyde (Sigma-Aldrich, France) for 20 minutes at room temperature, and tartrate-resistant acid phosphatase (TRAP) activity was detected by enzyme histochemistry to assess osteoclast differentiation. Cells were stained for acid phosphatase, using naphthol ASTR phosphate (Sigma-Aldrich, France) as a substrate in the

presence of 50 mM tartrate and hexazotized pararosaniline (Sigma-Aldrich, France) and counterstained with methyl green (Sigma-Aldrich, France). TRAP-positive cells with 3 or more nuclei were considered as osteoclasts and were counted under a bright-field microscope.

## Statistical analysis

All values were plotted as dots, and the mean ± SEM was calculated and plotted as bars. Statistical analyses were performed using GraphPad Prism 10 (GraphPad Software). Outliers were removed after applying the ROUT test. Statistical analysis was performed using t-test, two-way and three-way analysis of variance (ANOVA). Statistical significance was set at $*p < 0.05$; $**p < 0.01$, $***p < 0.001$, $****p < 0.0001$.

## Results

### Increased secretion of IL-1β in the *Sh3bp2* cherubism mouse model

We analyzed TNFα and IL-1β serum levels by Milliplex® assay in *Sh3bp2* KI mice compared to *Sh3bp2* WT mice and confirmed the previously reported increase of TNFα in *Sh3bp2* KI mice (Fig 1A). In addition, we found a significant increase in IL-1β serum levels in 10-week-old *Sh3bp2* KI mice (Fig 1B). This observation prompted us to explore the contribution of the inflammasome to the cherubism phenotype in this mouse model. We generated *Sh3bp2* KI *Caspase-1* KO double mutants by crossing heterozygotes in order to analyze the effect of inflammasome impairment on the cherubism phenotype, namely the bone-loss and inflammatory phenotypes. We also attempted to assess serum IL-18 levels in these mice, but, unfortunately, we were not able to detect it.

### Effects of Caspase-1 deficiency on morphological features of the *Sh3bp2* cherubism mouse model

The first obvious feature of the *Sh3bp2* KI cherubism model is the closed eyelid phenotype [8]. The Caspase-1 deficiency did not alter this feature in *Sh3bp2* KI;*Cas1* KO in both males

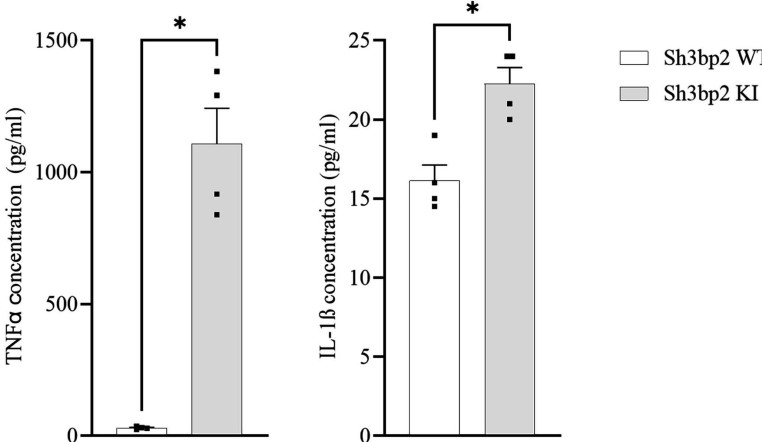

**Fig 1. TNFα and IL-1β serum levels in *Sh3bp2* KI mice at 10 weeks.** A. At 10 weeks, serum TNFα is significantly increased in *Sh3bp2* KI mice compare to WT mice. B. At 10 weeks, serum IL-1β is significantly increased in *Sh3bp2* KI mice compare to WT mice. Values are presented as dots (n = 4) and mean ± SEM. Statistical analysis was performed by t-test. Statistical significance was set at $*p < 0.05$.

and females (Fig 2A and C). Similarly, lack of Caspase-1 in *Sh3bp2* KI mice did not alter the lower body weight observed in *Sh3bp2* KI male mice or the smaller size of *Sh3bp2* KI male mice (Fig 2B). The same observations apply to Caspase-1-deficient *Sh3bp2* KI female mice (Fig 2D).

## Effects of Caspase-1 deficiency on the inflammatory phenotype of the *Sh3bp2* cherubism mouse model

The cherubism phenotype in *Sh3bp2* KI mice is characterized by elevated serum TNFα levels and the presence of inflammatory lesions in the liver [8]. We analyzed the effects of Caspase-1 deficiency on these features in *Sh3bp2* KI mice. Caspase-1 deficiency did not prevent the appearance of the liver inflammatory lesions in *Sh3bp2* KI male mice (Fig 3A and B). In *Sh3bp2* KI;*Cas1* KO male mice, the serum level of TNFα was still elevated as in *Sh3bp2* KI male mice (Fig 3C). The same observations apply to Caspase-1-deficient *Sh3bp2* KI female mice (Fig 3D–F). The serum level of IL-1β was significantly decreased by the Caspase-1 deficiency in *Sh3bp2* KI females (Fig 3F) and showed a trend toward a decrease in *Sh3bp2* KI males (Fig 3C).

## Effects of Caspase-1 deficiency on the mandibular bone phenotype of the *Sh3bp2* cherubism mouse model

The cherubism bone phenotype in *Sh3bp2* KI mice is characterized by mandibular bone loss [8]. We therefore analyzed the effects of the Caspase-1 deficiency on the mandibular bone phenotype in *Sh3bp2* KI mice. Caspase-1 deficiency did not alter the mandibular bone loss observed in *Sh3bp2* KI mice, as shown by the unchanged BV/TV measurements in both male and female mice (Fig 4).

## Effects of Caspase-1 deficiency on the systemic bone phenotype of the *Sh3bp2* cherubism mouse model

The cherubism bone phenotype in *Sh3bp2* KI mice is also characterized by systemic bone loss and increased calvarial resorption [8]. We next analyzed the effect of Caspase-1 deficiency on the bone phenotype in the *Sh3bp2* KI mice. Caspase-1 deficiency did not modify the lower BMD observed in *Sh3bp2* KI mice, either at the total whole-body level (Fig 5A and B), the femur level (Fig 5B), or the vertebral level (S1A Fig). Similarly, Caspase-1 deficiency did not alter the femoral bone loss in *Sh3bp2* KI;*Cas1* KO male mice (Fig 5), either at the femoral trabecular compartment or at the cortical level (Fig 5C, D, E and F). The same observations were made for the vertebrae (S1 Fig). Similar observations apply to *Sh3bp2* KI,*Cas1* KO female mice (Figs 5G–L and S2). To further our understanding of the effect of Caspase-1 deficiency on femur BV/TV according to the sex of the animal, we performed a 3-way ANOVA (S3 Fig). This analysis did not reveal any differences related to the Caspase-1 deficiency between males and females; however, it did confirm the effect of sex (p > 0.0001) and the effect of *Sh3bp2* KI (p > 0.0001) on femur BV/TV. Similarly, the calvarial bone resorption observed in *Sh3bp2* KI mice was not rescued by the lack of Caspase-1 in *Sh3bp2* KI;*Cas1* KO male mice (Fig 6A and B). Similar observations were made in the case of *Sh3bp2* KI;*Cas1* KO female mice (Fig 6C and D).

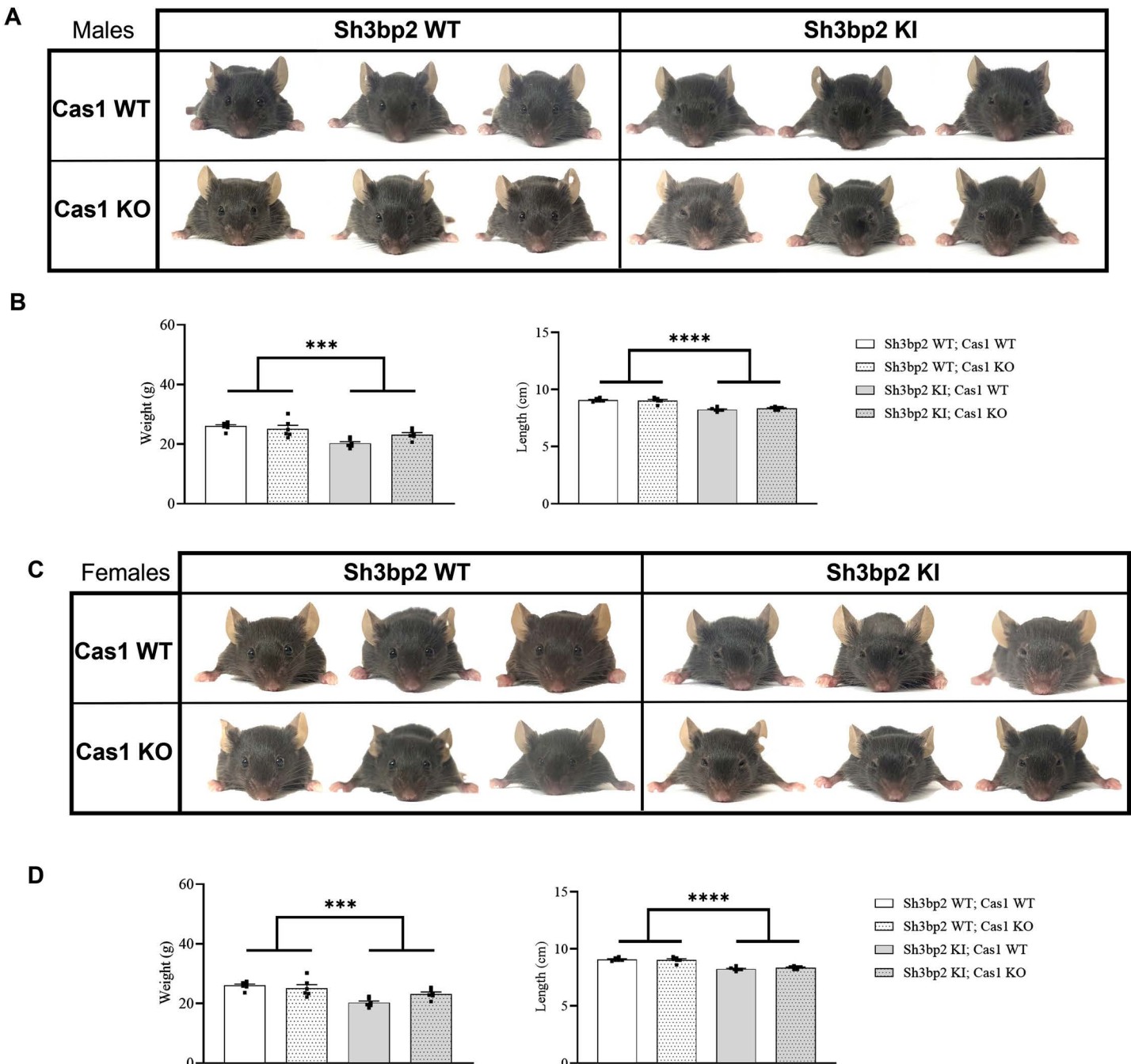

**Fig 2. Morphological features of *Sh3bp2;Cas1* mice at 10 weeks.** A. Facial appearance of *Sh3bp2;Cas1* male mice at 10 weeks of age (3/group). B. Body weight (left) and snout-to-tail length (right) of male mice. Deficiency of Caspase-1 in *Sh3bp2* KI mice does not rescue the reduced weight or the smaller size of males (n = 6/group). C. Facial appearance of *Sh3bp2;Cas1* female mice at 10 weeks of age (3/group). D. Body weight (left) and snout-to-tail length (right) of female mice. Deficiency of Caspase-1 in the *Sh3bp2* KI mice does not rescue the reduced weight or the smaller size of females (n = 6/group). Values are presented as dots and mean ± SEM. Statistical analysis was performed by two-way ANOVA. Statistical significance was set at ***p < 0.001, ****p < 0.0001.

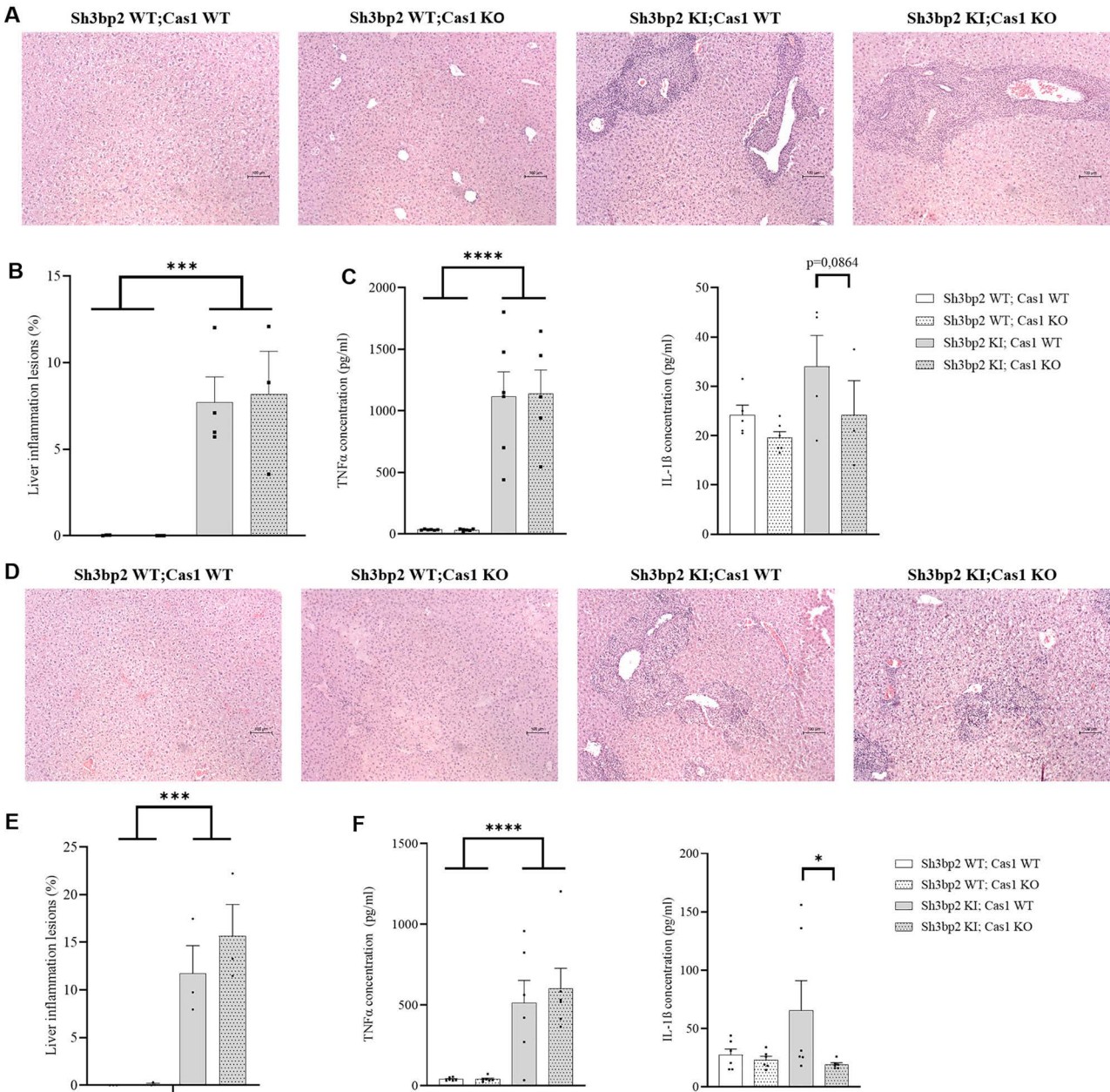

**Fig 3. Inflammatory phenotype of *Sh3bp2KI; Cas1KO* mice at 10 weeks.** A. Representative images of HE-stained liver sections of male mice showing inflammatory lesions (scale bar: 100 μm). B. Quantification of inflammatory lesion surface area as a percentage of the total area for each genotype (n = 3-4/group). C. Serum concentrations of TNFα and IL-1β in male mice of each genotype at 10 weeks of age (n = 6/group). D. Representative images of HE-stained liver sections of female mice showing inflammatory lesions (scale bar: 100 μm). E. Quantification of inflammatory lesion surface area as a percentage of the total area for each genotype (n = 3-4/group). F. Serum concentrations of TNFα and IL-1β in female mice of each genotype at 10 weeks of age (n = 6/group). Values are presented as dots and mean ± SEM. Statistical analysis was performed by two-way ANOVA. Statistical significance was set at * p < 0.05, ***p < 0.001, ****p < 0.0001.

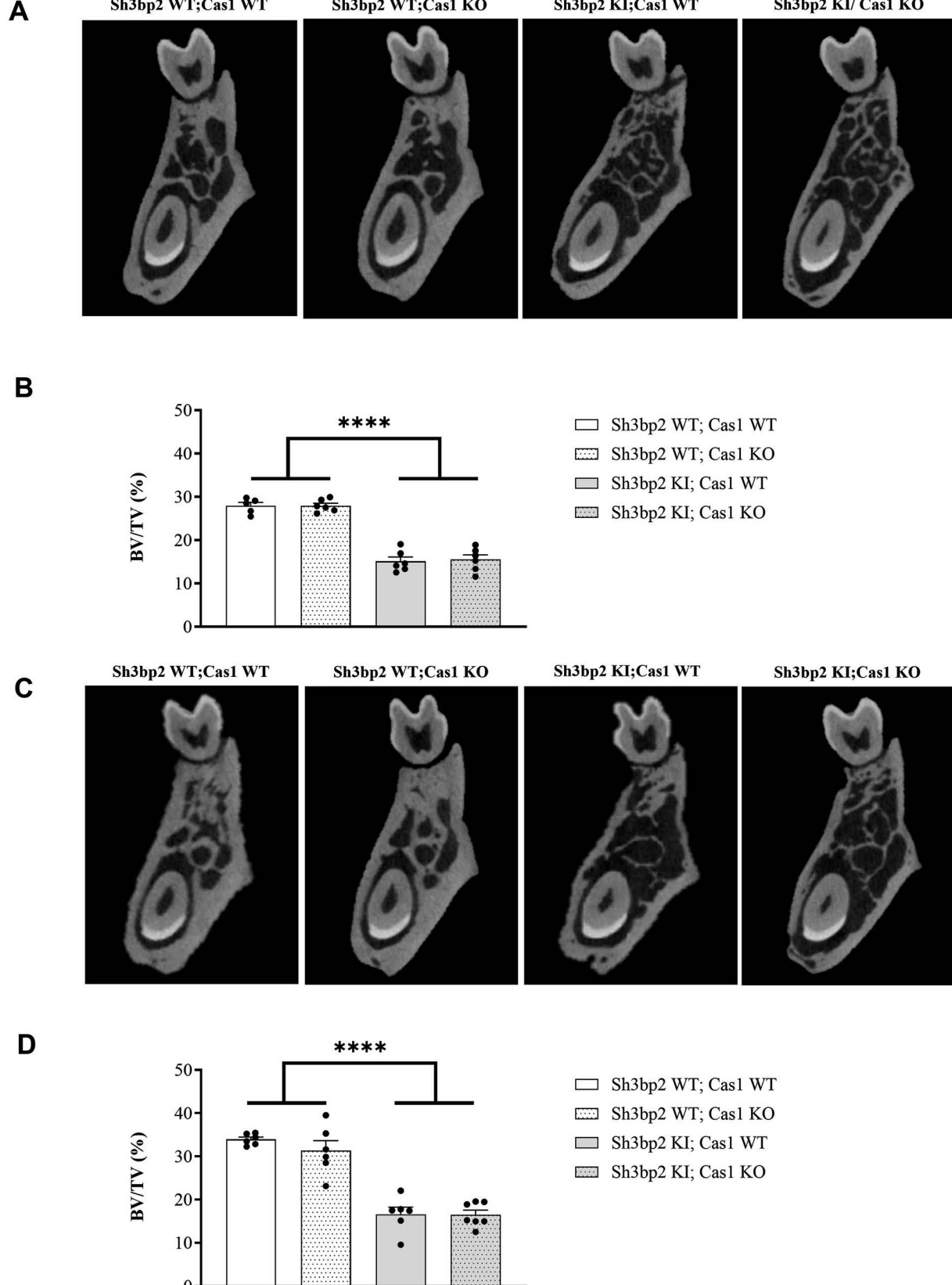

**Fig 4. Mandibular bone phenotype of *Sh3bp2; Cas1* mice at 10 weeks.** A. Representative images of the male mandibular bone in the furcation area for each genotype at 10 weeks. B. Bone volume/tissue volume (BV/TV) analysis for each genotype at 10 weeks of age. C.

Representative images of the female mandibular bone in the furcation area for each genotype at 10 weeks of age. D. BV/TV analysis for each genotype at 10 weeks of age. Values are presented as dots and mean ± SEM. Statistical analysis was performed by two-way ANOVA. Statistical significance was set at ****$p < 0.0001$.

### Effects of Caspase-1 deficiency on osteoclast differentiation in the *Sh3bp2* cherubism mouse model

Since *Sh3bp2* KI mice exhibit high osteoclast differentiation [8], we evaluated osteoclast differentiation in splenocyte cultures from the 4 mouse groups, and observed that the absence of Caspase-1 did not decrease the osteoclast differentiation observed in *Sh3bp2* KI male mice (Fig 7A and B). We also counted the number of TRAP-positive osteoclasts present in femur sections. We found that *Sh3bp2* KI male mice have an increased number of such cells and that this number is still high in the *Sh3bp2* KI;*Cas1* KO male mice (Fig 7C and D). The same results were observed in *Sh3bp2* KI;*Cas1* KO female mice (Fig 7E–H).

## Discussion

Cherubism is a rare genetic disorder caused mainly by gain-of-function mutations in the *SH3BP2* gene, resulting in osteolysis of the jaws [1]. To gain insight into the pathogenesis of the disease, mouse models have been generated using a knock-in strategy [7,8]. Most studies on cherubism in mice have been performed in the P416R KI model generated by Ueki and colleagues [8], mostly at 10 weeks of age. Here, we used the G418R KI model generated by our colleagues [7] and also performed our analyses at 10 weeks. In both models, mouse cherubism is characterized by systemic bone loss and inflammatory phenotypes mediated by elevated levels of TNFα [7,8]. More specifically, the cherubism bone-loss phenotype is characterized by jaw bone-loss, systemic bone loss observed in the femur, and an increase in calvarial bone erosion in *Sh3bp2* KI mice [8,11,18–23]. The mouse cherubism inflammatory phenotype is characterized by an increase in TNFα serum levels and the presence of inflammatory lesions in the liver of *Sh3bp2* KI mice [8,11,18–23]. Here, we characterized the consequences of *Caspase-1* deletion in *Sh3bp2* KI mice, using exactly the same methods as in previously published studies of mouse cherubism [8,11,18–23], adding only the analysis of the vertebral bone phenotype.

Cherubism has been described as a sterile autoinflammatory disease [28]. Among the cytokines involved in this type of pathology, IL-1β has been described as a mediator of sterile autoinflammation [12,13]. Furthermore, IL-1β is reported to be a stimulator of osteoclast differentiation [29]. In addition, previous studies suggested that IL-1β levels may be increased in *Sh3bp2* KI mice [10,11]. To support this point, we described a cherubism patient with a systemic bone-loss phenotype associated with elevated levels of IL-1β [5]. In the present study, we aimed to disentangle the impact of IL-1β in a *Sh3bp2* KI mouse model of cherubism and determine whether IL-1β could be a novel therapeutic target to rescue the bone phenotype. Since the NRLP3 inflammasome is responsible for IL-1β production through the action of Caspase-1, we targeted IL-1β production by deleting *Caspase-1* in *Sh3bp2* KI mice in an attempt to rescue the cherubism bone phenotype. First, we investigated the inflammatory status of *Sh3bp2* KI mice. We confirmed the elevated level of TNFα [7,8] and demonstrated for the first time that serum IL-1β is elevated in a cherubism mouse model. We then investigated the consequences of *Caspase-1* deletion on the cherubism bone and inflammatory phenotypes in the *Sh3bp2* KI model. Although we observed a significant decrease in IL-1β serum levels

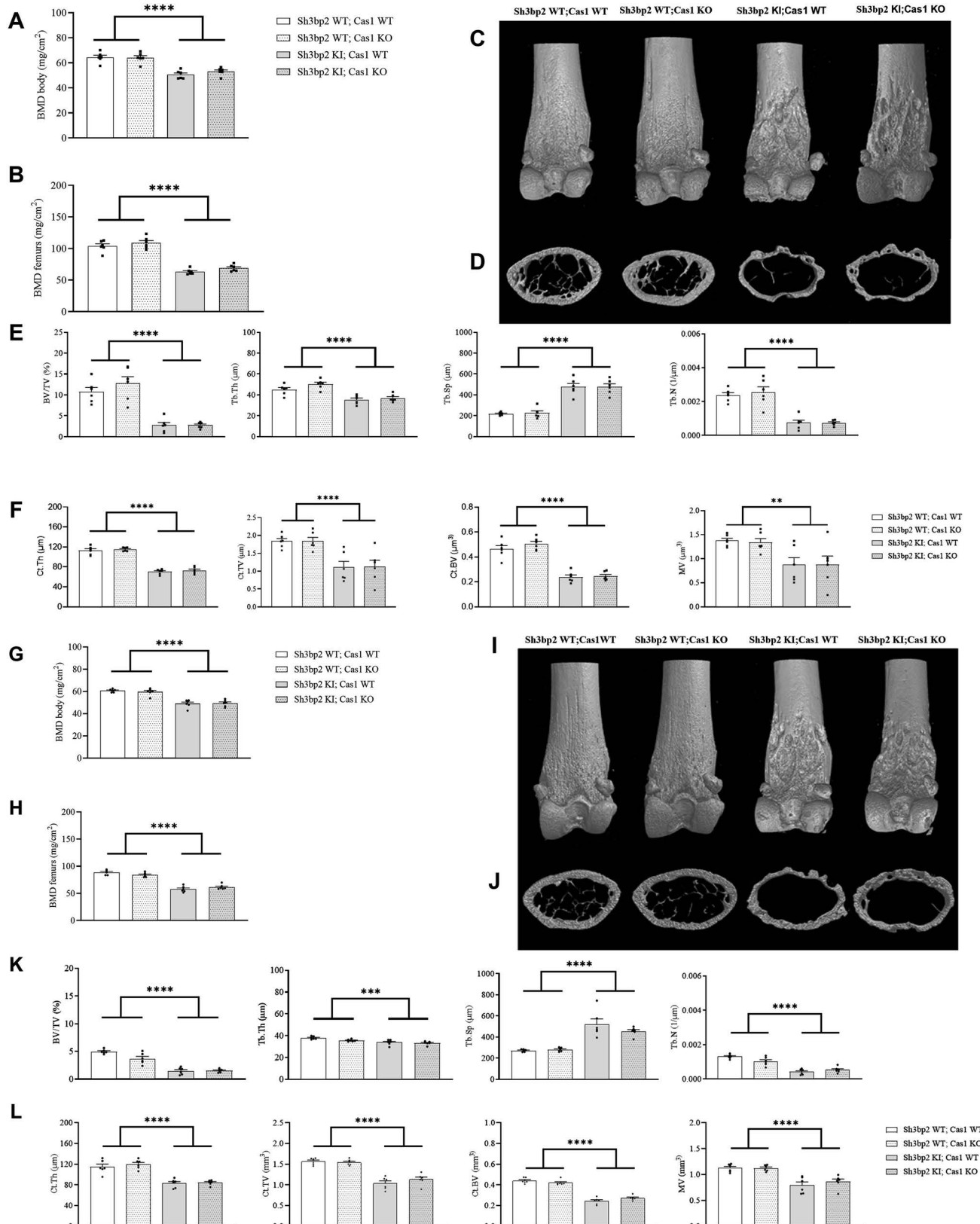

**Fig 5. Systemic bone phenotype of *Sh3bp2;Cas1* mice at 10 weeks.** A–F. Systemic bone phenotype of *Sh3bp2;Cas1* male mice and controls at 10 weeks of age. A. Body Bone Mineral Density (BMD) for each genotype. B. Femur BMD for each genotype. C. Representative 3D μCT femur reconstructions

showing osteolytic lesions. D. Representative 3D μCT coronal femur reconstructions for each genotype. E. Microarchitecture analysis of trabecular parameters (n = 6/group). F. Microarchitecture analysis of cortical parameters (n = 6/group). G–L. Systemic bone phenotype of *Sh3bp2;Cas1* female mice and controls at 10 weeks of age. G. Body BMD for each genotype. H. Femur BMD for each genotype. I. Representative 3D μCT femur reconstructions showing osteolytic lesions. J. Representative 3D μCT coronal femur reconstructions for each genotype. K. Microarchitecture analysis of trabecular parameters (n = 6/group). (BV/TV = Bone volume/Tissue volume; Tb.Th = trabecular thickness; Tb.Sp = trabecular separation; Tb.N = trabecular number). L. Microarchitecture analysis of cortical parameters (n = 6/group). (Ct.Th = cortical thickness; Ct.TV = cortical tissue volume; Ct.BV = cortical bone volume; MV = medullary volume). Values are presented as dots and mean ± SEM. Statistical analysis was performed by two-way ANOVA. Statistical significance was set at **p < 0.01, ***p < 0.001, ****p < 0.0001.

in females (and a trend in males), overall, deletion of *Caspase-1* in the *Sh3bp2* KI mouse model did not rescue either the bone or the inflammatory phenotypes. Indeed, we did not observe any improvement in the various bone parameters that we studied (BMD, bone microarchitecture, osteoclast number and differentiation) in either female or male mice. Taken together, these results seem to indicate that our hypothesis of NLRP3/ Caspase-1 involvement in the pathogenesis of cherubism is incorrect. Moreover, although the increased serum IL-1β level appears to be a consequence of the *Sh3bp2* KI mutation (as it is not observed in WT littermates), the elevated serum IL-1β does not seem to be instrumental in the actual cherubism phenotypes. Although the absence of Caspase-1 was associated with a decrease (or trend) in serum IL-1β levels, the absence did not completely abolish IL-1β levels. These data might suggest that the elevated levels of IL-1β have another source(s), possibly involving the non-canonical inflammasome pathway and other caspases. Alternatively, IL-1β may not be instrumental in the pathogenesis of cherubism and its upregulation only reflects the hypersensitivity of macrophages [10]. We must note that even *Sh3bp2 WT;Cas1 WT* mice have a relatively high basal level of IL-1β. This could be a sign of chronic inflammation in relation to our animal facility conditions (conventional and not Specific Pathogen Free (SPF)). Nevertheless, this study allowed us to develop a robust and powerful tool to evaluate calvarial bone resorption. This tool could be useful for analyzing bone erosion in other anatomical sites such as the cortical erosion of the femur described previously [8]. We also showed for the first time, at least to our knowledge, that *Sh3bp2* KI vertebrae are also affected by the cherubism bone-loss phenotype. Finally, we showed that unlike Caspase-3 and -7 [30,31], *Caspase-1* deletion does not affect bone. This may reflect the different basal activity levels of apoptotic caspases (Caspase-3 and -7) versus inflammatory caspases (Caspase-1) [13].

In a previous study performed in *Sh3Bp2* KI mice under germ-free conditions, inflammation was assessed by measuring the pro-inflammatory cytokine TNFα [10]. These axenic mice showed a significant reduction in TNFα compared to heteroxenic *Sh3Bp2* KI mice raised under SPF conditions. These data highlight an effect of the microbiota on inflammation and its consequences on the bone-loss phenotype in the cherubism mouse model. This raises the question of how to analyze the inflammation associated with *Sh3bp2* mutation in these mice. To avoid any confounding factors, these mice may need to be reared and tested under germ-free conditions.

In conclusion, the elevated serum level of IL-1β in our cherubism mouse model is not fully explained by the action of Caspase-1. Indeed, deletion of *Caspase-1* did not restore the bone and inflammatory phenotypes. Therefore, further studies are needed to explore the potential mechanisms underlined by the increased IL-1β, which could involve non-canonical inflammasome pathways.

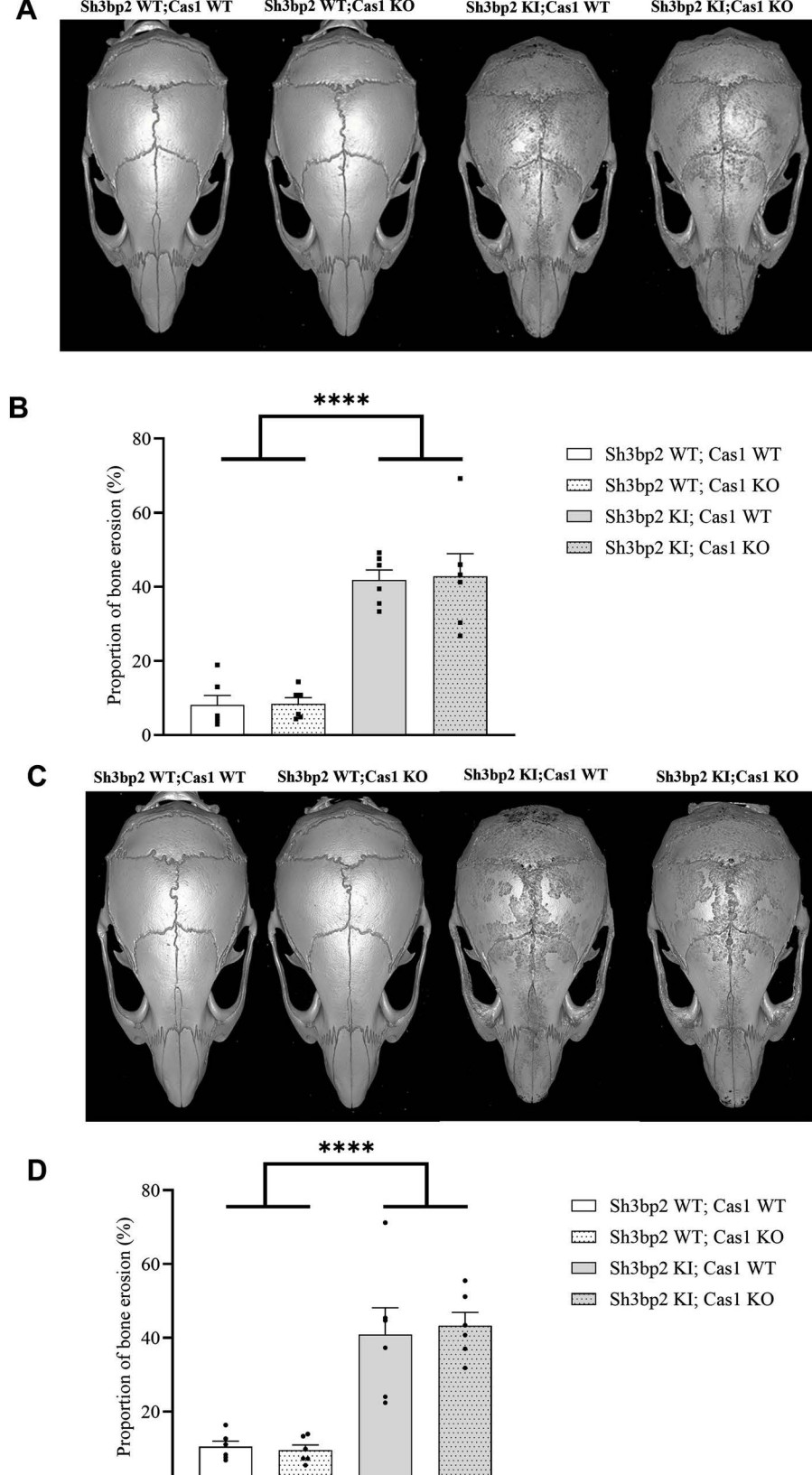

**Fig 6. Calvarial bone erosion of *Sh3bp2;Cas1* mice at 10 weeks.** A. Representative 3D μCT reconstructions of skulls from each genotype showing multiple osteolytic lesions in males at 10 weeks of age. B. Quantification of calvarial

bone erosion for each genotype. The ratio (%) of bone erosion to the total calvarial bone area (6 mm × 6 mm) was calculated for each genotype (n = 6/group). C. Representative 3D μCT reconstructions of skulls showing multiple osteolytic lesions in females at 10 weeks of age. D. Quantification of calvarial bone erosion for each genotype. The ratio (%) of bone erosion to the total calvarial bone area (6 mm × 6 mm) was calculated for each genotype (n = 6/group). Values are presented as dots and mean ± SEM. Statistical analysis was performed by two-way ANOVA. Statistical significance was set at ****p < 0.0001.

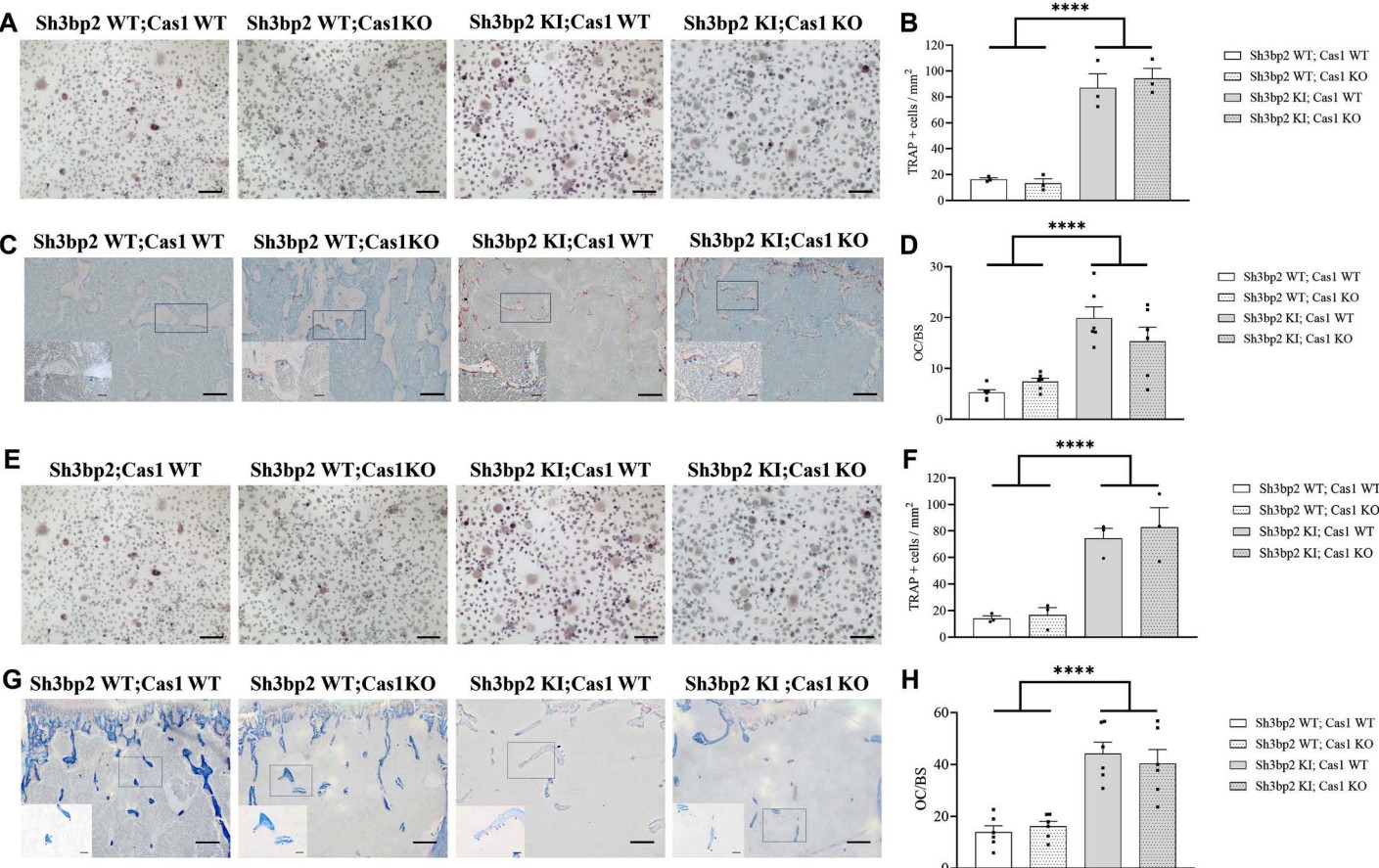

**Fig 7. *In vitro* osteoclast differentiation and osteoclast number in *Sh3bp2; Cas1* mice at 10 weeks.** A–D. *In vitro* osteoclast differentiation and osteoclast number in *Sh3bp2; Cas1* male mice at 10 weeks of age. A. Representative images of TRAP staining of primary osteoclast cultures from the spleen of *Sh3bp2WT;Cas1WT, Sh3bp2WT;Cas1KO, Sh3Bp2KI;Cas1WT, Sh3bp2KI;Cas1KO* mice after 14 days of culture with M-CSF and RANKL (scale bar: 50 μm). B. Number of TRAP-positive osteoclasts from primary osteoclast cultures of the 4 types of mice (n = 3/group). C. Representative images of TRAP-positive osteoclasts on undecalcified femur bone sections from the 4 types of mice (scale bar: 100 μm). D. Number of osteoclasts/bone surface area for each type of mouse (n = 6/group). E–H. *In vitro* osteoclast differentiation and osteoclast number in *Sh3bp2; Cas1* female mice at 10 weeks of age. E. Representative images of TRAP staining of primary osteoclast cultures from the spleen of *Sh3bp2WT;Cas1WT, Sh3bp2WT;Cas1KO, Sh3bp2KI;Cas1WT, Sh3bp2KI;Cas1KO* mice after 14 days of culture with M-CSF and RANKL (scale bar: 100 μm). F. Number of TRAP-positive osteoclasts from primary osteoclast cultures of the 4 types of mice (n = 3/group). G. Representative images of TRAP-positive osteoclasts on undecalcified femur bone sections from the 4 types of mice (scale bar: 100 μm). H. Number of osteoclasts/bone surface area for each type of mouse (n = 6/group). Values are presented as dots and mean ± SEM. Statistical analysis was performed by two-way ANOVA. Statistical significance was set at ****p < 0.0001.

## Supporting information

**S1 Fig. Vertebral bone phenotype of *Sh3bp2;Cas1* male mice at 10 weeks.** A. Male vertebral BMD for each genotype. B. Representative 3D μCT reconstructions of vertebrae showing multiple osteolytic lesions. C. Representative 3D μCT coronal reconstructions of male mouse vertebrae at 10 weeks of age for each genotype. D. Microarchitecture analysis of trabecular

parameters (BV/TV = Bone volume/Tissue volume; Tb.Th = trabecular thickness; Tb.Sp = trabecular separation; Tb.N = trabecular number) (n = 6/group). E. Microarchitecture analysis of cortical parameters (Ct.Th = cortical thickness; Ct.TV = cortical tissue volume; Ct.BV = cortical bone volume; MV = medullary volume) (n = 6/group). Values are presented as dots and mean ± SEM. Statistical analysis was performed by one two-way ANOVA. Statistical significance was set at * p < 0.05, **p < 0.01, ***p < 0.001, ****p < 0.0001.
(TIF)

**S2 Fig. Vertebral bone phenotype of *Sh3bp2;Cas1* females at 10 weeks.** A. Female vertebral BMD for each genotype. B. Representative 3D μCT reconstructions of vertebrae showing multiple osteolytic lesions. C. Representative 3D coronal μCT reconstructions of female mouse vertebrae at 10 weeks of age for each genotype. D. Microarchitecture analysis of trabecular parameters (BV/TV = Bone volume/Tissue volume; Tb.Th = trabecular thickness; Tb.Sp = trabecular separation; Tb.N = trabecular number) (n = 6/group). (C) Microarchitecture analysis of cortical parameters (Ct.Th = cortical thickness; Ct.TV = cortical tissue volume; Ct.BV = cortical bone volume; MV = medullary volume) (n = 6/group). Values are presented as dots and mean ± SEM. Statistical analysis was performed by two-way ANOVA. Statistical significance was set at ***p < 0.001, ****p < 0.0001.
(TIF)

**S3 Fig. Femur BV/TV analyses show no significant differences between sexes independent of Caspase-1 deletion at 10 weeks.** Femur BV/TV was measured in 10-week-old male and female mice for each genotype (n = 6/group). Values are presented as dots and mean ± SEM. Statistical analysis was performed using three-way ANOVA. Statistical significance was set at ****p < 0.0001 (male BV/TV vs. female BV/TV).
(TIF)

## Acknowledgments

The authors want to thank the IMRB Imaging Facility, the CHIC platform, the Viggo Petersen Animal facility and the BIOSCAR staff for their technical assistance.

## Author contributions

**Conceptualization:** Xavier Decrouy, Marcel Deckert, Amélie E. Coudert, François Brial.

**Data curation:** Xavier Decrouy, Amélie E. Coudert, François Brial.

**Formal analysis:** Xavier Decrouy, Amélie E. Coudert, François Brial.

**Funding acquisition:** Martine Cohen-Solal, François Brial.

**Investigation:** Badre-Victor Rabhi, Sylvie Thomasseau, Xavier Decrouy, Amélie E. Coudert, François Brial.

**Methodology:** Xavier Decrouy, Amélie E. Coudert, François Brial.

**Project administration:** Amélie E. Coudert, François Brial.

**Resources:** Amélie E. Coudert, François Brial.

**Supervision:** Amélie E. Coudert, François Brial.

**Validation:** Amélie E. Coudert, François Brial.

**Visualization:** Amélie E. Coudert, François Brial.

**Writing – original draft:** Amélie E. Coudert, François Brial.

**Writing – review & editing:** Martine Cohen-Solal, Amélie E. Coudert, François Brial.

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
