## [Decision Letter · Decision Letter 0]

9 Oct 2024

PONE-D-24-29696The bone phenotype associated with cherubism is independent of the inflammasome activation in mousePLOS ONE

Dear Dr. Coudert,

Thank you for submitting your manuscript to PLOS ONE. After careful consideration, we feel that it has merit but does not fully meet PLOS ONE’s publication criteria as it currently stands. Therefore, we invite you to submit a revised version of the manuscript that addresses the points raised during the review process.

 **The authors are required to address the following points during the revision.****1) Evaluate the phenotype of other skeletal sites besides jaw bones****2) Perform data analysis to evaluate sex-depenent effects of Sh3bp3 KI mutation****3) Perfrom ANOVA to confirm conclusions drawn based on T-test****4) Provide legible, high resolution figures****5) Revise text to elliminate misinterpretation of data and conclusions not supported by experimental data****6) Careful editing of the manuscript to correct spelling and grammar.  ** **In addition to the above required changes, the authors are also recommended to provide an indepth discussion for the lack of skeletal phenotype in the mutant mice.**

We look forward to receiving your revised manuscript.

Kind regards,

Subburaman Mohan

Academic Editor

PLOS ONE

**Journal Requirements:**

Reviewers' comments:

Reviewer's Responses to Questions

**Comments to the Author**

1. Is the manuscript technically sound, and do the data support the conclusions?

Reviewer #1: No

Reviewer #2: Partly

Reviewer #3: No

2. Has the statistical analysis been performed appropriately and rigorously? 

Reviewer #1: Yes

Reviewer #2: Yes

Reviewer #3: No

3. Have the authors made all data underlying the findings in their manuscript fully available?

Reviewer #1: No

Reviewer #2: Yes

Reviewer #3: Yes

4. Is the manuscript presented in an intelligible fashion and written in standard English?

Reviewer #1: No

Reviewer #2: No

Reviewer #3: No

5. Review Comments to the Author

**Reviewer #1:**  The Figures in PDF version of this manuscript are pixelated and hard to recognize, especially Figure 4, it is totally black. Therefore it is impossible to judge scientific merits of the work reported in this manuscript.

**Reviewer #2:**  This study focused on the involvement of IL-1 β in the cherubism pathogenesis, and the goal was to demonstrate that knocking down the expression of caspase 1 (Cas1) in Sh3bp2 knock-in (Sh3bp2 KI) mice will reverse the effect of cherubism on bone phenotype. For this purpose, they used Sh3bp2 Knock-In (Sh3bp2 KI) and Caspase1 knockout (Cas1 KO) mouse models. However, the results did not show any change in bone phenotype or any other parameters in the double mutant Sh3bp2 KI,Cas1 KO mice. Cherubism is a rare pediatric disease characterized by jaw osteolysis. Beginning in early childhood, both the lower jaw (the mandible) and the upper jaw (the maxilla) become enlarged as bone is replaced with painless, cyst-like growths. These growths give the cheeks a swollen, rounded appearance and often interfere with normal tooth development. In some people the condition is so mild that it may not be noticeable, while other cases are severe enough to cause problems with vision, breathing, speech, and swallowing.

- This is the first study that analyzed the phenotype of this Cherubism mutation, G418R KI, in mice. However, there is a major problem with this study. While TNF level in Sh3bp2-KI mice was almost-10-fold greater compared to WT mice, the level of Il-1beta in Sh3bp2-KI mice did not change much from WT mice (Fig. 1), and the authors did not explain the reason why the study focused on IL-1beta instead of TNF, and the graph in fig. 1 does not show if there is a significant difference between the level of Il-1beta in WT and Sh3bp2-KI mice.

- In Fig. 6.A. and Sup. Fig 8. Representative pictures of TRAP staining of primary osteoclast cultures from spleen from different genotypes. There are very few multinucleated Trap positive cells in WT spleen cells after 14 days treatment with MCSF and RANKL, and it is not obvious if the large cells from mutants are multinucleated or not. The authors did not explain why they chose 14 days, while it is known that splenocytes need 3-4 days to form Trap positive cells. Then, on day 4, mononuclear TRAP-positive cells will start to fuse to form multinuclear TRAP-positive cells, and after 7 days, multinuclear TRAP-positive cells may go through apoptosis.

- The paper is poorly written and needs to be revised and the English needs improvement.

**Reviewer #3: ** Cherubism, a rare dominant autosomal pediatric autoinflammatory disease characterized by a loss of bone mass with fibrous tissues replacement that was restricted to jaw bones. It is caused primarily by gain-of-function mutations on Sh3bp2 gene that led to increased secretion of TNFα. The authors found that 10 weeks-old Sh3bp2 knockin (KI) transgenic mice not only developed Cherubism but also showed elevated serum levels of IL-1β, which plays a central role in autoinflammatory diseases. This study sought to test the hypothesis that Caspase-1-mediated IL-1β secretion is involved in Cherubism by testing whether deletion of Caspase-1 would rescue the Cherubic bone phenotype in Sh3bp2 KI mutant mice. It was found that deletion of Caspase-1 did not reverse the abnormal bone phenotype of Sh3bp2 KI mice. The investigators concluded that the Caspase-1-mediated increased secretion of IL-1β played no roles in Cherubism and that the source of the elevated serum IL-1β may have come from Caspase-1-independent non-canonical inflammasome pathways.

This study has several significant deficiencies as summarized in the following:

1. With the exception the finding of an elevated serum IL-1β level in Sh3bp2 KI mice, the findings in this manuscript were not entirely new. In addition, the findings of this study in fact disproved the investigators’ original concept that Caspase-1-mediated released IL-1β has an important regulatory role in the abnormal bone and inflammatory phenotype in Sh3bp2 KI mutant mice. Thus, this study is of negative nature. Consequently, the clinical and scientific advances brought about by this study are incremental and only marginal.

2. In humans, the bone phenotype of Cherubism is largely restricted to the jaw bones, i.e., maxilla and mandible bones. Accordingly, although there was evidence in this and other studies that Sh3bp2 KI mutant mice also caused bone loss at other skeletal sites, it is disappointing that the investigators elected not to examine maxillomandibular bones and focused on other bone sites. Investigation of maxillomandibular bone could disclose additional key information as to whether the bone phenotype in these mutant mice was similar to that seen in humans. If the effects on maxillomandibular bones were not greater than other bone sites, it follows that expression of gain-of-function mutant of Sh3bp2 in mice might be different from that in humans. If so, it would suggest that this mutant mouse model might not be a suitable animal model for investigation of the pathophysiology of Cherubism in humans. As a result, the lack information on maxillomandibular bones represents a significant weakness of this manuscript.

3. An important deficiency of this manuscript is the lack of attention to the potential sex-dependent effects by not directly comparing the effects between male and female mice. Accordingly, it is important to document whether there were sex-dependent differences. [Some of the findings (see comment 5 below) might have shown significant sex-dependent differences]. In this regard, the data on male mice and those on female mice should be combined as a single Figure. This would then not only allow direct comparison between male and female mice but would also allow the determination whether there is a sex- and genotype-dependent interaction with two-ways ANOVA. Therefore, Fig. 2 and Suppl. Figs. 1 and 2 should be combined as a single figure. [Fig. 2A and Suppl. Fig. 1A were redundant, so Fig. 2A can be removed]. The same is true for Fig. 3 and Suppl. Fig. 3, Fig. 4 and Suppl. 4, Fig. 5 sand Suppl. Fig. 5, as well as Fig. 6 and Suppl. Fig. 7. Consequently, this deficiency renders the analysis of the data incomplete.

4. In all figures, the statistical analyses among the treatment groups combined the two Sh3bp2 WT groups (with or without Cas 1 KO) and the two Sh3bp2 KI groups (with or without Cas 1 KO) for comparison by t-test. This type of analyses completely ignored the contribution of Cas 1 deficiency and would have to assume that Cas 1 KO was not essential in the Sh3bp2 KI bone phenotype. This assumption completely contradicts the original hypothesis. Consequently, the statistical analysis of this study is inappropriate. All four groups should be separately compared and analyzed with ANOVA.

5. Many of the findings appeared to mis-interpreted, which led to erroneous conclusions in the manuscript. For example, there is no evidence in this manuscript supporting the conclusion that “these data might suggest that the increased production of IL-1β is involving the non-canonical inflammasome pathway and other caspases for this production”. On the contrary, the serum level of Sh3bp2 KI; Cas 1 KO double mutant mice was similar to that in Sh3bp2 KI WT mice (Fig. 3B and Suppl Fig. 3B). This piece of data would suggest that almost all the increased serum levels of IL-1β in both male and female Sh3bp2 KI mutant mice. Accordingly, there is no evidence for the increased serum IL-1β level in Sh3bp2 KI mutant mice was caused by any non-canonical inflammasome pathways other than the Caspase-1-dependent pathway. In addition, the increase in serum IL-1β level in female Sh3bp2 KI mutant mice was much greater than that in male Sh3bp2 KI mice. Similarly, the Cas 1 KO-related suppression in the increase in serum IL-1β level in female Sh3bp2 KI mutant was much larger than that in male mutant mice (Fig. 3B and Suppl. Fig. 3B). This would not support the conclusion of the investigators that there were no sex-dependent differences. Rather these findings suggest that there could be sex-dependent differences in these mutant mice. Finally, there was no direct measurement of osteoclastic resorption activities in this manuscript. Thus, no conclusion about bone resorption should be made.

6. The Discussion lacks in-depth discussion or potential explanation for the lack of effects of Caspase-1-mediated IL-1β activation in the bone phenotype of Cherubism in mice.

7. There were many typos, syntax errors and structural mistakes, as well as misuses of several English words throughout the manuscript (Please see specific comments for examples). The manuscript would benefit from careful editing for English by an English-speaking individual.

Specific comments:

1. Lines 1-2. The last word of the title should either be “mice” or “the mouse”. Also, it should be revised to indicate that it was independent of “the Caspase-1-dependent” inflammasome activation to be more specific because there are many types of inflammasomes.

2. Line 27. The word “invalidated” is vague and an inappropriate word. A better word would be “deleted” or “inactivated”. The same is true for many places throughout the manuscript.

3. Lines 30-32. The conclusion that “The source of the elevated IL-1β need more analysis but might come from the inflammasome non-canonical pathway, involving other caspases” was not tested. Thus, it should be a key conclusion and thereby should not be included in the Abstract.

4. Line 58. The word “scares” should be “scarce”.

5. Line 61. The word “funding” should be “founding”.

6. Line 68. The word “implies” should be “involves”.

7. Line 73. The sentence is not correct, because the study investigated the effects of deletion of Caspase-1 and not directly on IL-1β.

8. Lines 82-86. The breeding scheme for double mutants was not clear and should provide additional details. Genotyping methods for each mutant strain also need to be provided.

9. Line 91. Were the mice anesthetized or euthanized with ketamine and xylazine? In other words, were tissue samples and blood collected in anesthetized or euthanized animals? These procedures should be performed only in euthanized animals.

10. Line 97-98. 70% ethanol is not a good fixation agent for this study, since it can dehydrate specimens, dissolve proteins, lipids and alter collagen matrix structures. Therefore, it could confound the experimental results and interpretations.

11. Line 111. The word “exposition” is not an appropriate word. The more appropriate word may be “exposed for”.

12. Line 113. It should be “was used”.

13. Line 117. Please provide a rationale for why calvaria sutures were selected as the VOI for the µ-CT analyses.

14. Lines 127-131. At what site(s) were bone histomorphometry performed? Also, were all TRAP+ cells or only those on bone surface counted? This information is important because a large number of TRAP+ non osteoclastic cells were found in the marrow space. In addition, what does it mean “as usually done”? It is vague.

15. Line 134. It should be “for” 24 hr rather than “during” 24 hr.

16. Lines 137-141. Please explain what was “machine learning”, and how inflammation lesions were identified. Were clusters indices of inflammation? Also, it should be “The process was as the following”.

17. Line 152. All the four test groups were mutant mice. Accordingly, it is important to also include C57BL/6 wild-type mice as a control group to rule out any potential effects due to gene mutants in the Ch3bp2 KI WT and/or the Cas1 WT mice.

18. Line 156. Did you mean “the cell number in cell suspension was counted”?

19. Line 176. It states that two-way analysis of variance was performed. However, results of two-way ANOVA were not presented in this manuscript.

20. Line 182. It states that “increase of TNFα serum level in the Sh3bp2 KI mice is associated to an increase of the IL-1β serum level”. However, Figure 1 did not show an association between the two cytokines. In fact, there is no evidence that the levels of these two cytokines in the KI mice were associated to each other. Accordingly, the conclusion of an association is incorrect. They could be independent effects.

21. Lines 183-187. The canonical Caspase-1-mediated inflammasome activation pathway would also cause an increase in secretion of IL-18. Therefore, to confirm an involvement of the Caspase-1 mediated canonical inflammasome activation, the investigators may also wish to measure the serum level of IL-18.

22. Lines 189, 210, and 224. “Caspase 1 deficiency” is not appropriate than “Caspase 1 absence”.

23. Lines 194-195. What do the investigators mean when they stated that “the absence of the Caspase 1 in the Sh3bp2 KI mice did not modify the lower weight and height observed in the Sh3bp2 KI male mice or the smaller size observed in the Sh3bp2 KI male mice”? This sentence is unclear. Also, Fig. 2B did not measure height?

24. Line 204. It should be “the serum level of TNFα was still elevated …”.

25. Line 208. It should be “decrease”.

26. Lines 212. A rationale for focusing systemic bone loss phenotype and not evaluating the jaw bones, i.e., maxilla and mandible bones, (which is the major bone phenotype in humans) should be provided.

27. Lines 218-219. Please provide a rationale for why studies on vertebrae were done only in male mice and not female mice.

28. Lines 224-232. The bone resorption activity is the most important osteoclastic parameter. Increased TRAP+ cell number does not necessarily indicate an increased osteoclastic resorption activity. Therefore, why in vitro bone resorption activity (e.g., resorption pit formation assay) was not performed in Fig. 6.

29. Lines 234-238. This section is not related to the results or any figures or tables. Therefore, it does not belong to the Results section. It may belong to the Discussion section.

30. Lines 270-271. As indicated above, there is no evidence for the increased serum IL-1β level in Sh3bp2 KI mutant mice was caused by any non-canonical inflammasome pathways other than the Caspase-1-dependent pathway.

31. Lines 274-275. This reviewer does not understand the conclusion about “the relatively high basal level of IL-1β could be the sign of chronic inflammation in relation to our animal facility conditions”. Please elaborate.

32. Line 276. This reviewer also fails to understand the conclusion about that “this study allowed us to develop a robust and powerful tool to evaluate the calvarial bone resorption”.

33. Line 291. As described in above, the conclusion that the increased IL-1β serum level is not explained by the action of Caspase-1 in the cherubism mouse model” is incorrect.

34. Legend of Fig. 1. The “10 weeks” was stated three times. It is redundant. One time is sufficient. Also, the symbols **, ***, and **** are not shown in the figure. These symbols should be removed from the legend as the inclusion of which adds confusion.

35. Fig. 2B. Symbols should be added to indicate that Sh3bp2 KI significantly reduced the body weight and the body length. In addition, it appears that Cas 1 KO reverse the Sh3bp2 KI-dependent reduction in body weight. Similarly, the symbols * and ** should be removed from the figure legend to avoid confusion.

36. Legend of Fig. 3. “Dosages” should be “Concentrations or Levels”. Also, please remove * and ** from the legend as they were shown in the figure. To be consistent, the P=0.0864 may be stated in the legend instead on the figure.

37. The sub-labeling of legend of Fig. 4 is not consistent with what is shown in the figure. For example: B is not representative femur 3D reconstruction. C is not a representative femur 3D coronal µ-CT reconstruction; etc. Also, the symbols of * and *** should be removed from the legend as they were not shown in the figure.

38. Legend of Fig. 5. Please remove symbol description *, **, and *** be removed from the figure legend.

39. Fig. 6. It would be helpful for the reading to include arrows to indicate the osteoclasts on bone surface on panel C. Also, please remove symbols *, **, and *** from the legend as they were not shown on the figure.

6. PLOS authors have the option to publish the peer review history of their article (what does this mean? ). If published, this will include your full peer review and any attached files.

**Do you want your identity to be public for this peer review?** For information about this choice, including consent withdrawal, please see our Privacy Policy .

Reviewer #1: No

Reviewer #2: No

Reviewer #3: No

---

## [Author Response · Author response to Decision Letter 1]

24 Nov 2024

We would like to express our sincere gratitude to the reviewers for their valuable comments.

You will find a point-by-point response to all the feedback we received on our paper.

The authors are required to address the following points during the revision.

1) Evaluate the phenotype of other skeletal sites besides jaw bones

We have added the analysis of bone loss in the mandible. It was conducted as previously described by Yoshimoto et al. for example. We have added the analysis protocol in the materials and methods section, and add a paragraph in the results section.

2) Perform data analysis to evaluate sex-dependent effects of Sh3bp2 KI mutation

We conducted such analysis in order to answer the comment of the reviewer #3. However, comparing males and females does not make scientific sense as the females have by nature a lower bone density. In the bone field, males and females are always analyzed separately.

3) Perfrom ANOVA to confirm conclusions drawn based on T-test

In the materials and methods, we described the two-way ANOVA we conducted. The mention to a one-way ANOVA was a mistake copy and paste in all the figures.

4) Provide legible, high resolution figures

We modified the quality of the figures.

5) Revise text to eliminate misinterpretation of data and conclusions not supported by experimental data

We modified the discussion section.

6) Careful editing of the manuscript to correct spelling and grammar.

The manuscript was edited for spelling and grammar.

In addition to the above required changes, the authors are also recommended to provide an indepth discussion for the lack of skeletal phenotype in the mutant mice.

We improved the discussion in this sense.

Reviewer #1: The Figures in PDF version of this manuscript are pixelated and hard to recognize, especially Figure 4, it is totally black.

Therefore it is impossible to judge scientific merits of the work reported in this manuscript.

The figures were downloaded in a better quality especially in the pdf generated by PLOS One.

Reviewer #2: This study focused on the involvement of IL-1 β in the cherubism pathogenesis, and the goal was to demonstrate that knocking down the expression of caspase 1 (Cas1) in Sh3bp2 knock-in (Sh3bp2 KI) mice will reverse the effect of cherubism on bone phenotype. For this purpose, they used Sh3bp2 Knock-In (Sh3bp2 KI) and Caspase1 knockout (Cas1 KO) mouse models. However, the results did not show any change in bone phenotype or any other parameters in the double mutant Sh3bp2 KI,Cas1 KO mice. Cherubism is a rare pediatric disease characterized by jaw osteolysis. Beginning in early childhood, both the lower jaw (the mandible) and the upper jaw (the maxilla) become enlarged as bone is replaced with painless, cyst-like growths. These growths give the cheeks a swollen, rounded appearance and often interfere with normal tooth development. In some people the condition is so mild that it may not be noticeable, while other cases are severe enough to cause problems with vision, breathing, speech, and swallowing.

- This is the first study that analyzed the phenotype of this Cherubism mutation, G418R KI, in mice. However, there is a major problem with this study. While TNF level in Sh3bp2-KI mice was almost-10-fold greater compared to WT mice, the level of Il-1beta in Sh3bp2-KI mice did not change much from WT mice (Fig. 1), and the authors did not explain the reason why the study focused on IL-1beta instead of TNF, and the graph in fig. 1 does not show if there is a significant difference between the level of Il-1beta in WT and Sh3bp2-KI mice.

We did not focus on TNFα because it has extensively been done by Ueki et al in their original paper describing for the first time the Sh3bp2 KI mouse as a cherubism mouse model and the central role of TNFα role in the development of the cherubism phenotype in mouse. We wanted to explore the potential involvement of IL-1β in the mouse cherubism phenotype as some patients have increased IL-1β serum levels.

- In Fig. 6.A. and Sup. Fig 8. Representative pictures of TRAP staining of primary osteoclast cultures from spleen from different genotypes. There are very few multinucleated Trap positive cells in WT spleen cells after 14 days treatment with MCSF and RANKL, and it is not obvious if the large cells from mutants are multinucleated or not. The authors did not explain why they chose 14 days, while it is known that splenocytes need 3-4 days to form Trap positive cells. Then, on day 4, mononuclear TRAP-positive cells will start to fuse to form multinuclear TRAP-positive cells, and after 7 days, multinuclear TRAP-positive cells may go through apoptosis.

We use this protocol of osteoclast differentiation in our laboratory for quite some time now. There are as many osteoclast differentiation protocols as laboratory working in osteoclasts. Our protocol might not be the most efficient but is the working one in our hands.

- The paper is poorly written and needs to be revised and the English needs improvement.

The paper was edited for English by an English native spoken person.

Reviewer #3: Cherubism, a rare dominant autosomal pediatric autoinflammatory disease characterized by a loss of bone mass with fibrous tissues replacement that was restricted to jaw bones. It is caused primarily by gain-of-function mutations on Sh3bp2 gene that led to increased secretion of TNFα. The authors found that 10 weeks-old Sh3bp2 knockin (KI) transgenic mice not only developed Cherubism but IL-1β, which plays a central role in autoinflammatory diseases. This study sought to test the hypothesis that Caspase-1-mediated IL-1β secretion is involved in Cherubism by testing whether deletion of Caspase-1 would rescue the Cherubic bone phenotype in Sh3bp2 KI mutant mice. It was found that deletion of Caspase-1 did not reverse the abnormal bone phenotype of Sh3bp2 KI mice. The investigators concluded that the Caspase-1-mediated increased secretion of IL-1β played no roles in Cherubism and that the source of the elevated serum IL-1β may have come from Caspase-1-independent non-canonical inflammasome pathways.

This study has several significant deficiencies as summarized in the following:

1. With the exception the finding of an elevated serum IL-1β level in Sh3bp2 KI mice, the findings in this manuscript were not entirely new. In addition, the findings of this study in fact disproved the investigators’ original concept that Caspase-1-mediated released IL-1β has an important regulatory role in the abnormal bone and inflammatory phenotype in Sh3bp2 KI mutant mice. Thus, this study is of negative nature. Consequently, the clinical and scientific advances brought about by this study are incremental and only marginal.

We demonstrated for the first time that 1. the IL-1 β serum level is elevated in the Sh3bp2 KI mouse, 2. this elevated serum level is not instrumental in the cherubism phenotype in mice, 3 and not only involved the NLRP3 inflammasome as the caspase-1 invalidation only blunt and not prevent the IL1 β serum level increase, 4. the mouse cherubism bone loss is also observed in the vertebrae. We agree that our study is negative in nature but we also added some new data on the cherubism phenotype.

2. In humans, the bone phenotype of Cherubism is largely restricted to the jaw bones, i.e., maxilla and mandible bones. Accordingly, although there was evidence in this and other studies that Sh3bp2 KI mutant mice also caused bone loss at other skeletal sites, it is disappointing that the investigators elected not to examine maxillomandibular bones and focused on other bone sites. Investigation of maxillomandibular bone could disclose additional key information as to whether the bone phenotype in these mutant mice was similar to that seen in humans. If the effects on maxillomandibular bones were not greater than other bone sites, it follows that expression of gain-of function mutant of Sh3bp2 in mice might be different from that in humans. If so, it would suggest that this mutant mouse model might not be a suitable animal model for investigation of the pathophysiology of Cherubism in humans. As a result, the lack information on maxillomandibular bones represents a significant weakness of this manuscript.

We have added the data concerning the mandibular bone loss. As all model, this cherubism mouse model is not perfect but is useful and used for several years.

3. An important deficiency of this manuscript is the lack of attention to the potential sex-dependent effects by not directly comparing the effects between male and female mice. Accordingly, it is important to document whether there were sex-dependent differences. [Some of the findings (see comment 5 below) might have shown significant sex-dependent differences]. In this regard, the data on male mice and those on female mice should be combined as a single Figure. This would then not only allow direct comparison between male and female mice but would also allow the determination whether there is a sex- and genotype-dependent interaction with two-ways ANOVA.

Therefore, Fig. 2 and Suppl. Figs. 1 and 2 should be combined as a single figure. [Fig. 2A and Suppl. Fig. 1A were redundant, so Fig. 2A can be removed]. The same is true for Fig. 3 and Suppl. Fig. 3, Fig. 4 and Suppl. 4, Fig. 5 sand Suppl. Fig. 5, as well as Fig. 6 and Suppl. Fig. 7.

Consequently, this deficiency renders the analysis of the data incomplete.

Bone phenotype analysis is by essence never conducted comparing males and females as females have always lower bone density or BV/TV for instance. However, we did analyze the effect of the genotype and sex for the BV/TV as the reviewer can appreciate on the following figures.

The 3-way anova conducted to analyze those data gave this results :

Source of Variation % of total variation P value P value summary Significant?

Sex 2,191 0,3081 ns No

SH3BP2 4,609 0,1426 ns No

CASPASE 8,313 0,0518 ns No

Sex x SH3BP2 0,7539 0,5479 ns No

Sex x CASPASE 2,556 0,2715 ns No

SH3BP2 x CASPASE 4,420 0,1508 ns No

Sex x SH3BP2 x CASPASE 2,685 0,2599 ns No

ANOVA table SS (Type III) DF MS F (DFn, DFd) P value

Sex 685,9 1 685,9 F (1, 34) = 1,071 P=0,3081

SH3BP2 1443 1 1443 F (1, 34) = 2,253 P=0,1426

CASPASE 2603 1 2603 F (1, 34) = 4,063 P=0,0518

Sex x SH3BP2 236,0 1 236,0 F (1, 34) = 0,3685 P=0,5479

Sex x CASPASE 800,2 1 800,2 F (1, 34) = 1,249 P=0,2715

SH3BP2 x CASPASE 1384 1 1384 F (1, 34) = 2,161 P=0,1508

Sex x SH3BP2 x CASPASE 840,7 1 840,7 F (1, 34) = 1,313 P=0,2599

Residual 21778 34 640,5

So there is no effect of the 3 factors tested, and no significative difference between males and females

We conducted the same 3 – way ANOVA on the femur BV/TV as the reviewer can appreciate here :

As expected, there is a significative effect of the sex and of Sh3bp2, but not of Caspase 1 KO, as the data are the following :

Source of Variation % of total variation P value P value summary Significant?

Sex 24,66 <0,0001 **** Yes

SH3BP2 46,01 <0,0001 **** Yes

Caspase1 0,06125 0,6890 ns No

Sex x SH3BP2 12,38 <0,0001 **** Yes

Sex x Caspase1 0,8436 0,1424 ns No

SH3BP2 x Caspase1 0,03295 0,7690 ns No

Sex x SH3BP2 x Caspase1 0,9423 0,1217 ns No

ANOVA table SS DF MS F (DFn, DFd) P value

Sex 225,8 1 225,8 F (1, 40) = 65,45 P<0,0001

SH3BP2 421,3 1 421,3 F (1, 40) = 122,1 P<0,0001

Caspase1 0,5609 1 0,5609 F (1, 40) = 0,1626 P=0,6890

Sex x SH3BP2 113,4 1 113,4 F (1, 40) = 32,87 P<0,0001

Sex x Caspase1 7,725 1 7,725 F (1, 40) = 2,239 P=0,1424

SH3BP2 x Caspase1 0,3018 1 0,3018 F (1, 40) = 0,08746 P=0,7690

Sex x SH3BP2 x Caspase1 8,629 1 8,629 F (1, 40) = 2,501 P=0,1217

Residual 138,0 40 3,450

However, we modified the data presentation according to the reviewer’s comment and we presented in parallel the data obtained for the males and the females in the figures.

Obviously, if the reviewer whishes we always can present the male and the female mice in the same graph, however, it is not the standard way to do it in the bone field.

4. In all figures, the statistical analyses among the treatment groups combined the two Sh3bp2 WT groups (with or without Cas 1 KO) and the two Sh3bp2 KI groups (with or without Cas 1 KO) for comparison by t-test. This type of analyses completely ignored the contribution of Cas 1 deficiency and would have to assume that Cas 1 KO was not essential in the Sh3bp2 KI bone phenotype. This assumption completely contradicts the original hypothesis. Consequently, the statistical analysis of this study is inappropriate. All four groups should be separately compared and analyzed with ANOVA.

As stated in the Materials and Methods section, the analysis of the 4 groups were conducted by a two-way ANOVA. The mention of the one-way ANOVA in the figure legends was an inappropriate copy/paste. We apologize for this mistake.

5. Many of the findings appeared to mis-interpreted, which led to erroneous conclusions in the manuscript. For example, there is no evidence in this manuscript supporting the conclusion that “these data might suggest that the increased production of IL-1β is involving the non-canonical inflammasome pathway and other caspases for this production”. On the contrary, the serum level of Sh3bp2 KI; Cas 1 KO double mutant mice was similar to that in Sh3bp2 KI WT mice (Fig. 3B and Suppl Fig. 3B). This piece of data would suggest that almost all the increased serum levels of IL-1β in both male and female Sh3bp2 KI mutant mice. Accordingly, there is no evidence for the increased serum IL-1β level in Sh3bp2 KI mutant mice was caused by any non-canonical inflammasome pathways other than the Caspase-1-dependent pathway. In addition, the increase in serum IL-1β level in female Sh3bp2 KI mutant mice was much greater than that in male Sh3bp2 KI mice. Similarly, the Cas 1 KO-related suppression in the increase in serum IL-1β level in female Sh3bp2 KI mutant was much larger than that in male mutant mice (Fig. 3B and Suppl. Fig. 3B). This would not support the conclusion of the investigators that there were no sex-dependent differences. Rather these findings suggest that there could be sex-dependent differences in these mutant mice. Finally, there was no direct measurement of osteoclastic resorption activities in this manuscript. Thus, no conclusion about bone resorption should be made.

The reviewer is right, we have removed all statement about bone resorption as only osteoclast differentiation was analyzed.

6. The Discussion lacks in-depth discussion or potential explanation for the lack of effects of Caspase-1-mediated IL-1β activation in the bone phenotype of Cherubism in mice.

We have modified the discussion in accordance with the reviewer’s comment.

7. There were many typos, syntax errors and structural mistakes, as well as misuses of several English words throughout the manuscript (Please see specific comments for examples). The manuscript would benefit from careful editing for English by an English-speaking individual.

The manuscript has been edited for English by an English-speaking person.

Specific comments:

1. Lines 1-2. The last word of the title should either be “mice” or “the mouse”. Also, it should be revised to indicate that it was independent of “the Caspase-1-dependent” inflammasome activation to be more specific because there are many types of inflammasomes.

The sentence was modified according to the comment of the reviewer.

2. Line 27. The word “invalidated” is vague and an inappropriate word. A better word would be “deleted” or “inactivated”. The same is true for many places throughout the manuscript.

We removed ‘invalidated’ from the manuscript.

3. Lines 30-32. The conclusion that “The source of the elevated IL-1β need more analysis but might come from the inflammasome noncanonical pathway, involving other caspases” was not tested. Thus, it should be a key conclusion and thereby should not be included in the Abstract.

We remove this sentence from the Abstract.

4. Line 58. The word “scares” should be “scarce”.

We corrected it.

5. Line 61. The word “funding” should be “founding”.

We corrected it.

6. Line 68. The word “implies” should be “involves”.

We corrected it.

7. Line 73. The sentence is not correct, because the study investigated the effects of deletion of Caspase

---

## [Decision Letter · Decision Letter 1]

11 Dec 2024

PONE-D-24-29696R1The bone phenotype associated with cherubism is independent of the Caspase-1 dependent inflammasome activation in the mousePLOS ONE

Dear Dr. Coudert,

Thank you for submitting your revised manuscript to PLOS ONE. While the revisions made have improved the quality of the manuscript, there are still some pending concerns that remain to be addressed.  Please see attached reviewer comments.  We invite you to submit a revised version of the manuscript that addresses the points raised by the original reviewer on the revised version.  

We look forward to receiving your revised manuscript.

Kind regards,

Subburaman Mohan

Academic Editor

PLOS ONE

Reviewers' comments:

Reviewer's Responses to Questions

**Comments to the Author**

1. If the authors have adequately addressed your comments raised in a previous round of review and you feel that this manuscript is now acceptable for publication, you may indicate that here to bypass the “Comments to the Author” section, enter your conflict of interest statement in the “Confidential to Editor” section, and submit your "Accept" recommendation.

Reviewer #3: (No Response)

2. Is the manuscript technically sound, and do the data support the conclusions?

Reviewer #3: Yes

3. Has the statistical analysis been performed appropriately and rigorously? 

Reviewer #3: No

4. Have the authors made all data underlying the findings in their manuscript fully available?

Reviewer #3: Yes

5. Is the manuscript presented in an intelligible fashion and written in standard English?

Reviewer #3: No

6. Review Comments to the Author

Reviewer #3: The authors have satisfactorily addressed most of my previous concerns. However, the following remaining concerns may require additional attention.

1. The authors performed 3-way ANOVA analyses on BV/TV and showed no significant sex-dependent difference or interaction among Sh3bp2 KI, Caspase-1 KO, and sex. This information should be included in the revised manuscript. However, because the immediate downstream effect of Caspase-1 KO would be IL-1β. Specifically, as I have pointed out in my original review, the increase in serum IL-1β level in female Sh3bp2 KI mutants was much greater than that in male Sh3bp2 KI mutants. Similarly, the Cas 1 KO-related suppression in the increase in serum IL-1β level in female Sh3bp2 KI mutant was much larger than that in male mutant mice (Fig. 3). Accordingly, two-way or three-way ANOVA should also be performed on serum IL-1β to definitively rule out a sex-dependent effect. In addition, I also disagree with the authors that bone phenotype analysis is never conducted comparing males and females as females have lower bone density or BV/TV. On the contrary, many sex-dependent differences (for example by two-way ANOVA) in various bone phenotypes were analyzed, identified, and recently reported. I also disagree with the authors that presentation of data on males and females on the same figure is not the standard way to do in the bone field.

2. I still have issues with the authors’ decision to combine the two Sh3bp2 WT groups (with or without Cas 1 KO) as a single group and the two Sh3bp2 KI groups (with or without Cas 1 KO) as a single group for comparison in the various figures. This way of analyses assumed that the contribution of Cas 1 deficiency was not essential in the Sh3bp2 KI bone phenotype. Since the original hypothesis was to predict a significant role of Cas 1 deficiency on the bone phenotype, the combination of groups with and without Cas 1 KO is inappropriate. The authors have not provided a justification or rationale for their decision to combine the with and without Cas 1 KO as a single treatment for statistical analysis.

3. The word Cas 1“invalidation” is vague and inappropriate. The authors still choose to use Cas 1 invalidation to describe deletion of Cas 1 throughout the revised manuscript.

4. The primer listed in Table 1 needs additional important information. For example, please indicate which of them were 5’-primers and which were 3’-primers.

5. Although some people use 70% ethanol as fixation agent for bone histomorphometry, I strongly believe that it is not a good fixation method as 70% ethanol can dehydrate specimens, dissolve proteins, lipids and alter cellular and organelle as well as collagen matrix structures. Formalin would be a better fixation agent.

6. The authors did not address my previous question as to whether only TRAP+ cells on bone surface counted as osteoclasts. This information is important because large numbers of TRAP+ non-osteoclastic cells were found in the marrow space.

7. The authors have indicated that a new table summarizing the Caspase 1 KO bone phenotype data was added to the Discussion. However, no such table is found in the revised manuscript or in the Discussion section.

8. The resolution of all figures is still rather poor.

7. PLOS authors have the option to publish the peer review history of their article (what does this mean? ). If published, this will include your full peer review and any attached files.

**Do you want your identity to be public for this peer review?** For information about this choice, including consent withdrawal, please see our Privacy Policy .

Reviewer #3: No

---

## [Author Response · Author response to Decision Letter 2]

7 Jan 2025

We would like to express our sincere gratitude to the reviewers for their valuable comments.

You will find a point-by-point response to all the feedback we received on our paper.

The paper was edited for English by an English native spoken person.

Reviewer #3: The authors have satisfactorily addressed most of my previous concerns. However, the following remaining concerns may require additional attention.

1. The authors performed 3-way ANOVA analyses on BV/TV and showed no significant sex-dependent difference or interaction among Sh3bp2 KI, Caspase-1 KO, and sex. This information should be included in the revised manuscript.

We have included this information in supplemental data.

However, because the immediate downstream effect of Caspase-1 KO would be IL-1β. Specifically, as I have pointed out in my original review, the increase in serum IL-1β level in female Sh3bp2 KI mutants was much greater than that in male Sh3bp2 KI mutants. Similarly, the Cas 1 KO-related suppression in the increase in serum IL-1β level in female Sh3bp2 KI mutant was much larger than that in male mutant mice (Fig. 3). Accordingly, two-way or three-way ANOVA should also be performed on serum IL-1β to definitively rule out a sex-dependent effect.

We want to kindly remind the review that we had added such a graph in the previous answer. We share it here again.

The 3-way anova conducted to analyze those data gave this results :

Source of Variation % of total variation P value P value summary Significant?

Sex 2,191 0,3081 ns No

SH3BP2 4,609 0,1426 ns No

CASPASE 8,313 0,0518 ns No

Sex x SH3BP2 0,7539 0,5479 ns No

Sex x CASPASE 2,556 0,2715 ns No

SH3BP2 x CASPASE 4,420 0,1508 ns No

Sex x SH3BP2 x CASPASE 2,685 0,2599 ns No

ANOVA table SS (Type III) DF MS F (DFn, DFd) P value

Sex 685,9 1 685,9 F (1, 34) = 1,071 P=0,3081

SH3BP2 1443 1 1443 F (1, 34) = 2,253 P=0,1426

CASPASE 2603 1 2603 F (1, 34) = 4,063 P=0,0518

Sex x SH3BP2 236,0 1 236,0 F (1, 34) = 0,3685 P=0,5479

Sex x CASPASE 800,2 1 800,2 F (1, 34) = 1,249 P=0,2715

SH3BP2 x CASPASE 1384 1 1384 F (1, 34) = 2,161 P=0,1508

Sex x SH3BP2 x CASPASE 840,7 1 840,7 F (1, 34) = 1,313 P=0,2599

Residual 21778 34 640,5

So there is no effect of the 3 factors tested, and no significative difference between males and females

In addition, I also disagree with the authors that bone phenotype analysis is never conducted comparing males and females as females have lower bone density or BV/TV. On the contrary, many sex-dependent differences (for example by two-way ANOVA) in various bone phenotypes were analyzed, identified, and recently reported. I also disagree with the authors that presentation of data on males and females on the same figure is not the standard way to do in the bone field.

We think there is a misunderstanding. We have no trouble to present the males and females data in the same figure, as the reviewer can observe in the new version of the figures.

Concerning the analysis of males and females in the same graph, we agree with the reviewer, that it might have been recently reported that way. We just prefer not to, for simplification purposes.

2. I still have issues with the authors’ decision to combine the two Sh3bp2 WT groups (with or without Cas 1 KO) as a single group and the two Sh3bp2 KI groups (with or without Cas 1 KO) as a single group for comparison in the various figures. This way of analyses assumed that the contribution of Cas 1 deficiency was not essential in the Sh3bp2 KI bone phenotype. Since the original hypothesis was to predict a significant role of Cas 1 deficiency on the bone phenotype, the combination of groups with and without Cas 1 KO is inappropriate. The authors have not provided a justification or rationale for their decision to combine the with and without Cas 1 KO as a single treatment for statistical analysis.

We can understand the reviewer’s concerns about the way we conducted the statistical analysis of our data. However, we conducted 2-way ANOVA analysis to explore the effect of the Caspase-1 deletion and the SH3BP2 KI. We honestly don’t know how we could have done it differently. If the reviewer wants to suggest another way to analyze the data, we would be more than happy to discuss this issue further in order to satisfy the reviewer.

3. The word Cas 1“invalidation” is vague and inappropriate. The authors still choose to use Cas 1 invalidation to describe deletion of Cas 1 throughout the revised manuscript.

The reviewer is right. We didn’t pay enough attention. We apologize. We removed all the invalidation and replaced it by deletion.

4. The primer listed in Table 1 needs additional important information. For example, please indicate which of them were 5’-primers and which were 3’-primers.

The reviewer is right. We forgot to give these important information. It has been corrected.

5. Although some people use 70% ethanol as fixation agent for bone histomorphometry, I strongly believe that it is not a good fixation method as 70% ethanol can dehydrate specimens, dissolve proteins, lipids and alter cellular and organelle as well as collagen matrix structures. Formalin would be a better fixation agent.

We think that there is a misunderstanding regarding the use of 70% ethanol. We agree with the reviewer that 70% ethanol is not the best fixative agent. We use 70% for our histomorphometry analysis in MMA sections especially because our protocols were designed like that, but we use formalin when our samples are embedded in paraffin.

6. The authors did not address my previous question as to whether only TRAP+ cells on bone surface counted as osteoclasts. This information is important because large numbers of TRAP+ non-osteoclastic cells were found in the marrow space.

We thought we did in the material and methods. We modified the sentence to be clearer.

7. The authors have indicated that a new table summarizing the Caspase 1 KO bone phenotype data was added to the Discussion. However, no such table is found in the revised manuscript or in the Discussion section.

It was a mistake and we indeed didn’t add any table. We apologize.

8. The resolution of all figures is still rather poor.

The reviewer is right. We have noticed that the resolution of all figures in the generated PDF is still poor. However, this was out of our control. If you click on the figure, the quality is much improved.

---

## [Editor Report · Decision Letter 2]

22 Jan 2025

The bone phenotype associated with cherubism is independent of Caspase-1-dependent inflammasome activation in mouse.

PONE-D-24-29696R2

Dear Dr. Coudert,

We’re pleased to inform you that your manuscript has been judged scientifically suitable for publication and will be formally accepted for publication once it meets all outstanding technical requirements.

Kind regards,

Subburaman Mohan

Academic Editor

PLOS ONE
---

## [Editor Report · Acceptance letter]

PONE-D-24-29696R2

PLOS ONE

Dear Dr. Coudert,

I'm pleased to inform you that your manuscript has been deemed suitable for publication in PLOS ONE. Congratulations! Your manuscript is now being handed over to our production team.

Kind regards,

on behalf of

Dr. Subburaman Mohan

Academic Editor

PLOS ONE